# The curve not taken: Effects of COVID-19 international comparison news

**Seon-Woo Kim**[1,2]*, **Martina Santia**[1¤a], **Raymond J. Pingree**[1], **Ayla Oden**[1],
**Kirill Bryanov**[1¤b], **Jessica Wyers**[1]

1 Mass Communication, Louisiana State University, Baton Rouge, Louisiana, United States of America,
2 Analytics, Louisiana State University, Baton Rouge, Louisiana, United States of America

¤a Current address: S.I. Newhouse School of Public Communications, Syracuse University, Syracuse, New York, United States of America
¤b Current address: The Social and Cognitive Informatics Laboratory, Higher School of Economics, St. Petersburg, Russia

* kr.seonwoo@gmail.com

**Data Availability Statement:** https://doi.org/10.7910/DVN/RKCM19.

**Funding:** The author(s) received no specific funding for this work.

## Abstract

International news can inform people not only about what is happening in other countries, but also about how their own country could benefit from policies that have proved successful elsewhere. Specifically, international policy comparison news, or news that compares the policies of two or more countries on the same issue, is a potentially important but underutilized and understudied form of news content. We use an experiment to test effects of exposure to news comparing the COVID-19 pandemic policies of the U.S. versus South Korea, and find that this increases knowledge of policy differences between the two countries, support for adopting similar policies in the U.S., presidential blame for the severity of the pandemic in the U.S., and trust in health experts. On most outcomes, these effects did not vary across political party lines, a particularly encouraging result given the polarized nature of policy debates on this issue.

## Introduction

Disasters can be crucial moments for significant policy learning, but it is not a given that we learn valuable lessons that actually improve policy [1]. Because COVID-19 is affecting many countries simultaneously and because countries are addressing it with very different policies, the pandemic presents a particularly compelling opportunity for the public to learn from news that compares across countries. In other domains of policy such as gun control and health care, the U.S. public's perceptions of what is possible have arguably been limited by unfamiliarity with how these problems have been more successfully addressed in other countries.

South Korea's approach to the COVID-19 pandemic presents a particularly instructive comparison to the U.S. response. Their approach to testing and contact tracing allowed isolation measures to be focused on individuals with potential exposure [2–4]. In the U.S., such individually targeted measures weren't possible due to early failures in the availability of testing and continuing failures in its speed. As a result, the blunt instrument of widespread lockdowns (with their much higher economic impacts) was the only option. Familiarity only with the U.S. response might lead the public to mistakenly conclude that such costly measures were the only

**Competing interests:** The authors have declared
that no competing interests exist.

possible approach prior to vaccines, and that they remain so for further efforts against the pandemic or future pandemics. In other words, the U.S. may have learned an exactly wrong lesson about pandemics: that the only things that can be done prior to vaccine availability or in addition to vaccines have overwhelming economic costs and questionable effectiveness.

This study used an experiment to test effects of a news story comparing COVID-19 responses by South Korea to the U.S. on knowledge of policy differences between the two countries, support for similar policies used successfully in South Korea, presidential blame for the severity of the pandemic in the U.S., and trust in health experts. We compare this treatment to a control condition where only the disease outcomes of the same two countries are compared, as well as to another control condition that only addresses the current state of the pandemic in the U.S.. The experiment was fielded on May 1, 2020, a crucial date in the U.S. policy debate over reopening when many health experts were warning that testing and tracing capabilities were still insufficient to safely relax lockdowns [5–7].

## Policy learning and news

Democratic theory stresses the importance of citizen policy knowledge, meaning knowledge about the available options for addressing societal problems and the likely costs and consequences of each option [8, 9]. Although ideal levels of policy knowledge may be unrealistic, any increase in policy knowledge could increase the quality of public opinion [10, 11], help hold elected officials accountable for poor policy choices [12], and increase the efficiency and quality of representation [13, 14].

There is no shortage of depressing evidence that the public is poorly informed about and uninterested in policy [14–16], at least in ordinary times. However, disasters can sometimes be moments of dramatic, widespread policy learning [1, 17, 18]. Whether valuable lessons are learned from a disaster depends in large part on perceptions of causality and responsibility, and particularly how people believe the disaster might have played out differently with different policies [1, 19].

Such questions can be particularly difficult and contested in one-off isolated events such as hurricanes, tsunamis or major terrorist attacks that are the prototypical events studied in research on learning from disasters. Because the COVID-19 pandemic is occurring nearly simultaneously in nearly every country, with each country taking different policy approaches, such comparisons need not be made in purely hypothetical terms. Citizens may find it very helpful to consider how other countries responded differently and whether they had different outcomes. This leads to our interest in effects of news that draws such international policy comparisons about the COVID-19 pandemic, particularly comparisons to countries such as South Korea that have had greater success both in terms of controlling the pandemic and in terms of doing so at much lower economic costs.

Although past theoretical work has emphasized the importance of policymakers learning from other countries [19, 20], empirical research on effects of international news on the general public has focused much more on outcomes such as attitudes about other countries and foreign policy [21–24]. Note that also unlike other research on international news, we focus specifically on news that compares policies across countries. International policy comparison is also conceptually distinct from international outcome comparison news, in which only the results achieved by different countries are compared, not the differences in policies that may explain those different results. In the context of COVID-19 coverage, such outcome-only comparison news might give a false impression that the pandemic will soon fade on its own in one country because it did so in other countries.

Several decades of newsroom cost-cutting have led to major reductions in international newsgathering resources at U.S. news outlets [25] while also greatly reducing policy coverage and instead focusing on political competition and strategy [26, 27]. There are commendable efforts underway to reverse this trend such as the solutions journalism movement [28–30]. We believe that international policy comparison news is a particularly promising form of solutions journalism. While most news agencies may remain ill-equipped for providing either international news or policy news, public demand for this content is high, at least during the present crisis. Recent studies from the Pew Research Center [31] suggest that Americans want to learn and improve U.S. policies by reading and learning about the ways other countries have responded. Just 47 percent of Americans are confident in the U.S. government's response to COVID-19, whereas 84 percent believe the U.S. could learn from other countries. And there is observational evidence suggesting that news about COVID-19 might even help slow the virus itself: within China, early media coverage of COVID-19 has been linked to reduced spread [32].

Accordingly, we hypothesize that exposure to news comparing the U.S. response to another country which had more successful outcomes will increase knowledge of policy differences between the U.S. and that country, support for implementing successful policies in other countries, and increase the attribution of the Trump administration' responsibility for the COVID-19 crisis.

H1a: Exposure to international policy comparison news will increase knowledge of policy differences between the two countries.

H1b: Exposure to international policy comparison news will decrease the misunderstanding that South Korea shut down its economy to suppress its virus outbreak

H1c: Exposure to international policy comparison news will decrease the misunderstanding that the U.S. government's policies for COVID-19 were similar to South Korea's

H2: Exposure to international policy comparison news will increase support for policies successful in the comparison country

H3: Exposure to international policy comparison news will increase presidential blame for the severity of the U.S. COVID-19 crisis

We also expect international policy comparison news to increase trust in health experts in cases where it provides easily understood evidence backing up already-familiar policy recommendations of experts. In the present context, the policies that seem to have been the most successful in other countries are the same policies that health experts have been emphasizing for months (specifically, large-scale testing and contact tracing to allow individual isolation instead of broad lockdowns). At the time of this study, these expert recommendations had also been amplified by opposition to them from certain political elites who claimed that the desire for more testing was driven by political ulterior motives of making the pandemic look worse [33]. Thus, it seems likely that at the time of this study many people were already aware that some were advocating greatly increased testing, while perhaps not being aware exactly why this would be so helpful in addressing the pandemic. Accordingly, we expect that when people learn that policies experts had been advocating were already very successful in other countries, this may not only increase support for those policies as hypothesized above but may also increase trust in health experts. Such trust is an extremely important outcome in its own right because of its importance in leading to compliance with recommendations from health experts, a crucial variable influencing efficacy of a wide variety of public health interventions [4, 34].

H4: Exposure to international policy comparison news will increase trust in health experts

Contrary to what people often assume about the influence of news media, research typically finds relatively limited effects that differ substantially for different subsets of the audience [35, 36]. In some cases where corrective messages are perceived as threatening to prior beliefs tied to salient identities, it can actually backfire, strengthening the misperceptions they are designed to correct [37]. Thus, it is important to examine possible moderating variables that might condition the strength or direction of any of our predicted effects. In particular, we focus on party preference. In the U.S., beliefs and attitudes about COVID-19 have become polarized along party lines, perhaps in part because Democrats are more likely to trust the news media about COVID-19 and Republicans are more likely to trust government officials [38]. This suggests that party may play an important role, but it could condition effects in complex ways. For example, it is possible that Republicans may be less receptive to our treatment if they see it as an implicit criticism of their party, but it's also possible that they would be more receptive if their selective exposure to in-party media had led them to be less exposed to similar content in the past [39, 40].

RQ1: Will any of the above effects be moderated by party preference?

## Method

The study was approved by the Louisiana State University Institutional Review Board on Apr 19, 2020, approval #E12156. To test our hypotheses, we relied on an online survey experiment that manipulated exposure to one of three versions of a news story about COVID-19. To maximize believability of the stimulus, we embedded the treatment story in a website page identical to a story page from the Wall Street Journal's website. The Wall Street Journal was chosen because it is a newspaper with relatively high levels of trust regardless of partisanship. We used a three-level treatment factor (international policy comparison news: policy comparison, outcome comparison, and a control condition with no international comparison) drawn from a larger fully factorial design intended for other purposes beyond this study. Three other 2-level factors each manipulated the presence or absence of one of the stories in the sidebar within the same stimulus page. These non-hypothesized factors were included in the analyses for control purposes.

The experiment was conducted on May 1, 2020 when states were announcing plans to reopen and loosen restrictions while the country recorded more than 1 million cases and at least 63,000 deaths [41]. May 1 was a crucial date in the U.S. policy debate over reopening as many health experts warned the administration that testing and tracing capabilities were still insufficient for safely relaxing lockdowns across states [5, 6].

### Participants

We recruited a convenience sample of U.S. adults (final $N$ = 848; 59.0% male, 40.4% female, 0.6% other) from Amazon's Mechanical Turk (MTurk), an online platform where registered users opt to participate in studies for a nominal payment. Participants received $1.40 for completing the study. The sample size decision was based on available resources and a rule of thumb for media effects experiments of 50 participants per cell [42]. Retrospectively, we ran a power analysis based on an effect size of Cohen's f = .14 from a solutions journalism experiment [29] and .95 power using G*power application, resulting in a required sample size of 792. The retrospective power analysis confirmed that our sample size is appropriate. MTurk convenience samples are generally considered similar in utility to researchers as the undergraduate

student convenience samples widely used in experimental research, but have much greater demographic diversity [43]. Of the sample, 73.6% identified as White or Caucasian, 14.6% as Black or African American, 5.7% as Asian, 5.9% as Hispanic or Latino, and 1.3% indicated other ethnicities. In answer to the question "Which political party do you prefer?", 42.5% of participants indicated a preference for the Republican Party, 48.9% preferred the Democratic Party, and 8.6% indicated not even a slight preference for either party. 77.1% of participants had a four-year college degree with an average age of 37.92 years (SD = 11.80, min = 18, max = 82). Participants received $1.40 for completing the study.

## Procedures

MTurk participants were randomly assigned to read one of the three stimulus news stories that appeared to be a screenshot from the *Wall Street Journal* website. Each story was identical in length at 646 words. Each screenshot featured the news story under the *Wall Street Journal's* U.S. header section, and used the exact fonts and styles of a real *Wall Street Journal* story webpage. The newspaper website layout and its navigation elements, such as content categories, social sharing buttons, and stock tickers, were held constant across all conditions.

The policy comparison condition (N = 281) featured a short news story explaining why South Korea was successful in managing the COVID-19 outbreak. This story explained that South Korea quickly emerged as a model to emulate without locking down entire cities, shutting down its economy, or taking some of the authoritarian measures implemented by countries like China. The policy comparison condition explained how mass testing and tracing efforts allowed South Korea to contain the virus outbreak without having to suppress economic activities and without impacting its democratic system.

The outcome comparison condition (N = 291) featured another short news story mentioning South Korea's success at containing the spread of the virus without explaining how it managed the situation. This story basically reported the small number of cases in South Korea relative to the exponential rise of cases in the United States. Finally, the control condition featured an update on the COVID-19 emergency in the United States with no international comparison.

The control story (N = 276) merely indicated the state and scale of the emergency in the United States and around the world using data from the World Health Organization (WHO) and reports from Johns Hopkins University. Once participants read one of the three stories, they were instructed to complete a short follow-up survey used to assess our variables of interest. Each story included an identical graph illustrating the trajectory of new virus cases in the United States. A similar graph for South Korea was also included in the two stories that mentioned South Korea (i.e. it was not included in the control story that only focused on the U.S. response).

## Measures

Unless otherwise stated, all outcome variables were measured on a seven-point Likert-type scale ranging from "1" labeled as "strongly disagree" to "7" labeled as "strongly agree."

Knowledge of policy differences between the two countries was operationalized using three indicators: testing & tracing, lockdown misperception, and policy similarity misperception. These were each measured with a single item: "South Korea used widespread testing and tracking people's movements to isolate people who might be infected" (M = 5.56, SD = 1.29), "South Korea mostly shut down its economy to suppress its outbreak" (M = 4.44, SD = 1.92), and "The U.S. is currently doing as much as South Korea did when it succeeded in reducing its outbreak" (M = 3.88, SD = 2.00).

Support for policies successful in other countries ($M$ = 5.27, $SD$ = 1.08, Cronbach's alpha = .77) was an average of respondents' levels of agreement with a set of six closely related statements assessing support for testing, tracing, and related technology implemented by South Korea to reduce the spread of COVID-19. Participants were asked to indicate whether they agreed or disagreed with the following statements: "Tests for COVID-19 should be available to the general population," "There should be more drive-thru testing centers," "We should wait until we have more COVID-19 testing capability before re-opening the economy," "We should wait until we have a system to trace contracts of infected people before re-opening the economy," "Governments should create a phone app that allows people to report symptoms," and "The government should use cellular data to track citizens' potential exposure to COVID-19."

Presidential blame ($M$ = 4.62, $SD$ = 1.95) was a single item: "COVID-19 spread widely in the U.S. because of the Trump Administration."

Trust in health experts ($M$ = 5.68, $SD$ = 1.07, Cronbach's alpha = .82) was an average of responses to the following questions: "How often do you think scientists set aside personal political views and make fair decisions based on evidence?", "How often do you think doctors set aside personal political views and make fair decisions based on evidence?" and "How often do you think epidemiologists (i.e., scientists who study infectious diseases) set aside personal political views and make fair decisions based on evidence?". Responses ranged from "1" labeled as "never" to "7" labeled as "always."

## Results

To test our hypotheses, we used six analysis of variance (ANOVA) models, three for the three indicators of policy difference knowledge for testing & tracing (H1a), lockdown misperception (H1b), policy similarity misperception (H1c), one for support for policies successful in the comparison country (H2), one for presidential blame (H3), and one for trust in health experts (H4). The RQ1 test used the interactions between party preference and the treatment in each model. Each model included a three-level treatment factor (international policy comparison news: policy comparison, outcome comparison, and a control condition with no international comparison), a three-level party preference variable (Republican Party, Democratic Party, and not even a slight preference for either party). Other non-hypothesized factors were included in all analyses for control purposes. In addition, to remove outliers for our models, we used Cook's distance [44] with cutoff value, 4/n, as a general rule of thumb [45]. As a result, 33 observations of each model were deleted on average for final models. Excluding outliers did not substantively affect the results. Overall, it slightly increased the effect sizes. Fig 1 shows the treatment effects on each outcome by political identification.

H1a predicted that exposure to international policy comparison news will increase knowledge of policy differences. Regarding the ANOVA model for H1a, exposure to international policy comparison news significantly increased the understanding of South Korean testing and tracking system to isolate people who might be infected (F[2, 774] = 44.66, $p$ < .001, partial $\eta^2$ = .103). Post-hoc comparisons using HSD Tukey test showed that the policy comparison group ($M_p$ = 6.18, $SD_p$ = .88), the outcome comparison group ($M_o$ = 5.61, $SD_o$ = 1.04), and the control group ($M_c$ = 5.30, $SD_c$ = 1.10) were significantly different at $p$ < .05. This result supports H1a.

In answering H1b, exposure to international policy comparison news significantly decreased the misunderstanding that South Korea shut down its economy to suppress its virus outbreak (F[2, 781] = 24.72, $p$ < .001, partial $\eta^2$ = .060). As for H2a, HSD Tukey post hoc comparisons at $p$ <. 05 showed that the policy comparison group ($M_p$ = 3.74, $SD_p$ = 2.19) was different from the outcome comparison group ($M_o$ = 4.66, $SD_o$ = 1.69) and the control group ($M_c$

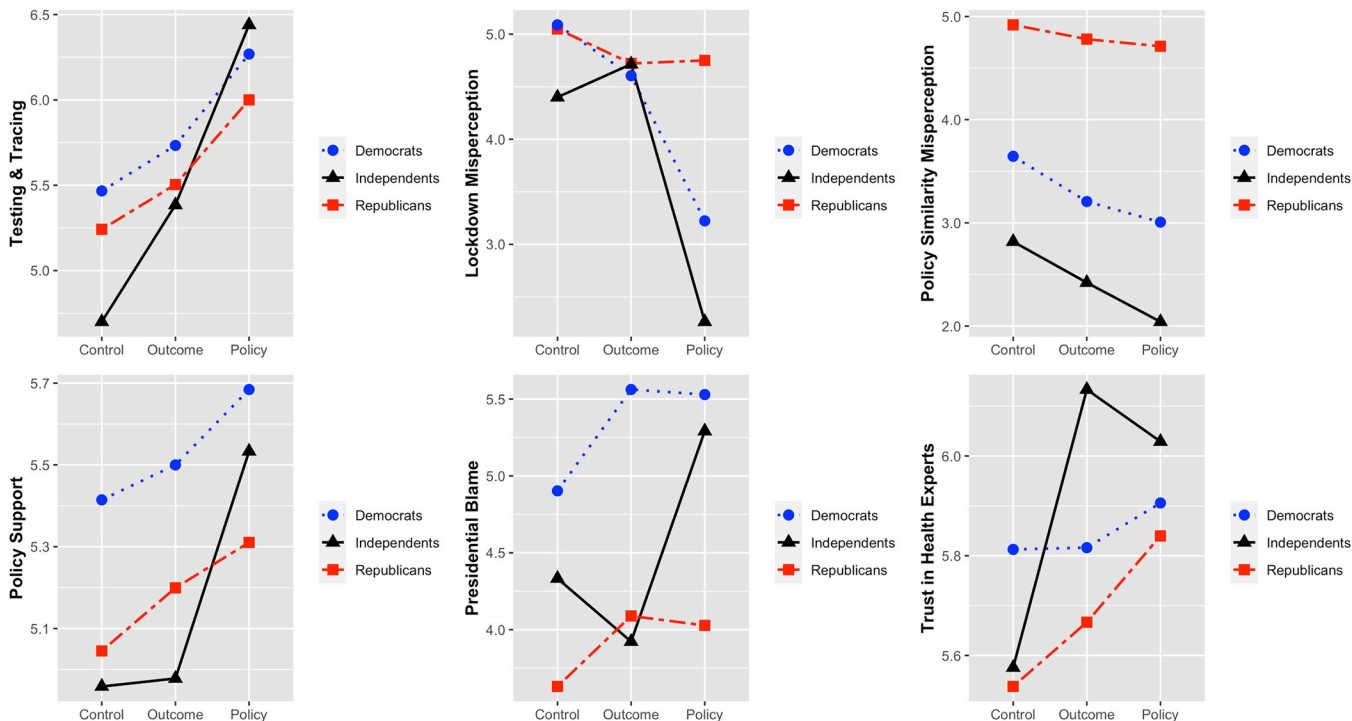

**Fig 1. Effect on knowledge of policy difference, support for policy, and trust toward health experts.**

= 5.02, $SD_c$ = 1.42). However, the outcome comparison group and the control group did not significantly differ. This result supports H1b.

After testing H1c, we found that exposure to international policy comparison news significantly decreased the misperception that the U.S. government's policies for COVID-19 were similar to South Korea's (F[2, 801] = 3.45. $p$ < .05, partial $\eta^2$ = .009). HSD Tukey tests showed that the policy comparison group ($M_p$ = 3.58, $SD_p$ = 2.04) perceived less similarity between the two countries' policies than the control group ($M_c$ = 4.16, $SD_c$ = 1.85). In contrast, the outcome comparison group ($M_o$ = 3.84, $SD_o$ = 2.02) did not differ from either the control group or the policy comparison group. This result supports H1c.

As predicted by H2, the ANOVA model revealed that exposure to international policy comparison news significantly increased support for policies successful in the comparison country (F[2, 777] = 5.38. $p$ < .01, partial $\eta^2$ = .014). Post hoc tests using HSD Tukey showed that the policy comparison group ($M_p$ = 5.53, $SD_p$ = 0.91) was more likely to support policies successful in South Korea compared to the outcome comparison group ($M_o$ = 5.34, $SD_o$ = 0.89) and the control group ($M_c$ = 5.22, $SD_c$ = 1.00). However, the outcome group did not differ from the control group. This result supports H2.

As predicted by H3, exposure to international policy comparison news significantly increased presidential blame for the severity of the U.S. COVID-19 crisis (F[2, 785] = 5.15, $p$ < .01, partial $\eta^2$ = .013). Tukey HSD post hoc tests revealed that the policy comparison group ($M_p$ = 4.91, $SD_p$ = 1.83) was more likely to blame the Trump administration for the COVID-19 crisis in the United States than the control group ($M_c$ = 4.28, $SD_c$ = 1.93). However, the outcome comparison group ($M_o$ = 4.82, $SD_o$ = 1.86) was not different from the policy comparison group and the control group. This result supports H3.

As predicted by H4, exposure to international policy comparison news significantly increased trust in health experts (F[2, 768] = 2.65, $p$ < .05, partial $\eta^2$ = .007). Post hoc

comparisons using Tukey HSD tests indicated that the policy comparison group ($M_p$ = 5.89, $SD_p$ = 0.80) was significantly different from the control group ($M_c$ = 5.67, $SD_c$ = 0.96). However, the outcome comparison group ($M_o$ = 5.77, $SD_o$ = 0.92) did not significantly differ from the policy comparison group and the control group. This result supports H4.

In answer to our RQ1 about party preference as a moderator, out of all six outcomes only the first two indicators of policy knowledge had significant interactions between party and the treatment. We conducted Tukey HSD post hoc analysis for the interaction of treatment and political party preference to follow up on those two significant interactions (see S1 Table for full ANOVA results). Each hypothesis model has 32 post hoc comparison pairs in the interaction term. However, our explanation focuses on the effect of international policy comparison news within the same political parties to determine whether political preference moderate the treatment effect on outcome variables. Thus, we only compare policy news treatment with each of the two other conditions within the same party preference (Democrat, Republican, or independent), for a total of six contrasts for each outcome.

For testing and tracing knowledge, despite the significant omnibus interaction term, the six contrasts of interest showed homogeneous effects. Specifically, the policy comparison group and control comparison group exhibit significant differences among Republicans (difference = 0.76, $p < .001$), Democrats (difference = 0.80, $p < .001$), and independents (difference = 1.71, $p < .001$). Similarly, the policy comparison group differed from the outcome group among Republicans (difference = 0.50, $p < .01$), Democrats (difference = 0.53, $p < .001$), and independents (difference = 1.09, $p < .05$).

In the model predicting the lockdown misperception, contrasts revealed that the effects of the treatment were among Democrats and independents, not Republicans. Specifically, the policy comparison group and control comparison group significantly differed among Democrats (difference = -1.92, $p < .001$), and independents (difference = -1.90, $p < .01$), but not among Republicans (difference = -0.27, n.s.). The policy comparison group and outcome group also significantly differed among Democrats (difference = -1.44, $p < .001$) and independents (difference = -2.24, $p < .01$) but not among Republicans (difference = 0.07, n.s.).

## Discussion

In sum, this study found that exposure to international policy comparison news improved understanding of policy differences, increased support for adopting policies similar to the comparison country, increased presidential blame for the severity of the problem domestically, and increased trust in relevant experts. These effects generally did not differ across party lines, with the exception of one of the three policy knowledge indicators, where learning among Republicans was not significant.

The effects on policy support and health expert trust are particularly compelling in hindsight given the timing of this study on May 1, 2020. This was a crucial moment in the U.S. COVID-19 pandemic when new cases had been in gradual decline for a month, resulting in debates about how and whether to re-open. Against the advice of prominent health experts who warned at the time that testing and tracing capacity was not yet sufficient [5–7], numerous states did re-open over the subsequent month, and new cases then began to surge again by mid-June, 2020.

Beyond the specific issue domain used here, these results suggest that international policy comparison is a promising form of solutions journalism worthy of further experimentation, both by media researchers and by media outlets. In contrast to the broader category of international news, which may primarily serve to influence knowledge of other countries and opinions about foreign policy, international policy comparison news can also foster more informed

opinions about domestic policy. It may also increase democratic accountability by providing comparisons that shed light on how elites could have done better. Although this experiment tested effects of this type of content in the COVID-19 pandemic, we see no reason to assume its potential benefits are limited to pandemics, health issues, or even to moments of crises. Ordinary domestic policy coverage on any issue might benefit from comparisons to the policies of other countries, potentially contributing to a healthy broadening of the range of policy options under debate. In other words, it is worth exploring as a strategy for addressing the problem of indexing, which is the tendency of news media to limit policy discussion to the range of views expressed by domestic political elites [46, 47].

We acknowledge this content is most likely rare in current practice. Both international news and policy news have been in decline for decades, driven in part by cost-cutting pressures [25–27]. This may explain the lack of past scholarly attention to the effects of international policy comparison news, although there has been research on effects of international news [22, 23] and policy news [26] separately. In our view, experimental research can best inform positive change by studying effects of what is needed, not only effects of what currently exists.

As with all experiments using a convenience sample, external validity is an important limitation. Future research should attempt to replicate these results with a representative sample. Caution is also warranted about generalizing beyond our stimuli to the broader categories of media content they are intended to represent. It is possible that the effectiveness of our particular treatment story relative to the two control stories is due to some idiosyncratic features of these stories and is not representative of the effects of international policy comparison news in general. Similarly, these effects may operate differently in other issue contexts, where reader interest is lower, or where the other country's problem or policy solutions can more easily be dismissed as incomparable. As with any experiment, the remedy to all of these external validity limitations is replication in future research. We hope other researchers will join us in exploring the effects of international policy comparison news.

## Supporting information

**S1 Table. ANOVA results summary.**
(PDF)

## Author Contributions

**Conceptualization:** Seon-Woo Kim, Martina Santia, Raymond J. Pingree.

**Data curation:** Seon-Woo Kim, Raymond J. Pingree, Kirill Bryanov, Jessica Wyers.

**Formal analysis:** Seon-Woo Kim.

**Investigation:** Seon-Woo Kim, Martina Santia, Raymond J. Pingree, Ayla Oden.

**Methodology:** Seon-Woo Kim, Raymond J. Pingree, Ayla Oden.

**Project administration:** Raymond J. Pingree.

**Software:** Seon-Woo Kim.

**Supervision:** Raymond J. Pingree.

**Validation:** Seon-Woo Kim, Raymond J. Pingree.

**Visualization:** Seon-Woo Kim.

**Writing – original draft:** Seon-Woo Kim, Martina Santia, Raymond J. Pingree, Ayla Oden, Kirill Bryanov, Jessica Wyers.

**Writing – review & editing:** Seon-Woo Kim, Martina Santia, Raymond J. Pingree, Ayla Oden, Kirill Bryanov, Jessica Wyers.

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
