## [Editor Report · Decision Letter 0]

24 Mar 2021

PONE-D-20-26385

The curve not taken: Effects of COVID-19 international comparison news

PLOS ONE

Dear Dr. Kim,

Thank you for submitting your manuscript to PLOS ONE. After careful consideration, we feel that it has merit but does not fully meet PLOS ONE’s publication criteria as it currently stands. Therefore, we invite you to submit a revised version of the manuscript that addresses the points raised during the review process.

We look forward to receiving your revised manuscript.

Kind regards,

Anima Nanda

Academic Editor

PLOS ONE

Journal Requirements:

2. Please ensure that you refer to Figure 1 in your text as, if accepted, production will need this reference to link the reader to the figure.

3.We note that Supporting Information S1, S2 and S3 in your submission contain copyrighted images. All PLOS content is published under the Creative Commons Attribution License (CC BY 4.0), which means that the manuscript, images, and Supporting Information files will be freely available online, and any third party is permitted to access, download, copy, distribute, and use these materials in any way, even commercially, with proper attribution. For more information, see our copyright guidelines: http://journals.plos.org/plosone/s/licenses-and-copyright.

a) You may seek permission from the original copyright holder of Supporting Information S1, S2 and S3 to publish the content specifically under the CC BY 4.0 license.

Additional Editor Comments:

Dear Author,

The article entitled “The curve not taken: Effects of COVID-19 international comparison news” is an interesting article. But there are few queries to be clarified.

Queries:

1) Is it possible to always address the international news and policy news simultaneously?

2) How far this article will help in improving policy?

3) Is the three-level treatment factor (international policy comparison news: policy comparison, outcome comparison, and a control condition with no international comparison) are significant to each other?

4) How these samples plays an vital role in external validity?

With regards

Dr. Nanda
---

## [Author Response · Author response to Decision Letter 0]

11 May 2021

We are delighted to be invited to revise our manuscript in light of the reviews. We appreciate the close reading of our manuscript and constructive feedback. We believe we have addressed all points raised and we believe these edits leave our manuscript much improved. Below we walk through the changes we made.

2. Please ensure that you refer to Figure 1 in your text as, if accepted, production will need this reference to link the reader to the figure.

We added the reference to Figure 1 in the first paragraph of the “Results” section as follows:

"The RQ test used the interactions between party preference and the treatment in each model (see Fig 1)."

3.We note that Supporting Information S1, S2 and S3 in your submission contain copyrighted images. 

We removed the copyrighted images from all stimuli and replaced them with equivalent images that we created.

Queries:

1) Is it possible to always address the international news and policy news simultaneously?

This is a good point and we believe that more theorizing and empirical work needs to be done to better distinguish between international news and policy news coverage. Even though the combination of international news and policy news is a promising form of news content, the media cover each news section separately (i.e., international news section & political news). Researchers have also studied them as independent research subjects. Thus, many potential effects remain to be seen. However, this study experimentally found strong evidence to support the benefits of international comparison news. In addition, depending on characteristics of issue and comparison characteristics, international policy comparison news effect would not work. We added it to discussion section further with highlighted marks. 

2) How far this article will help in improving policy?

We thank the reviewer for raising this point. In our manuscript, we argue that international comparison news coverage may increase democratic accountability by providing comparisons that shed light on how other countries address specific issues, such as the Covid-19 crisis. International comparison news coverage may also help political elites and institutions choose desirable policies while warning citizens against malpractice, injustice, and misdeeds. Informed citizenry, accurate attribution of policy evaluation to politicians, and trust in experts are all expected to improve the quality of domestic policies, potentially contributing to a healthy broadening of range of policy options under debate. We added the example of gun control policy in the discussion section to show how comparing domestic policy to the policies of other countries may prove advantageous for a variety of reasons. 

3) Is the three-level treatment factor (international policy comparison news: policy comparison, outcome comparison, and a control condition with no international comparison) are significant to each other?

We report the post-hoc analyses in the three-level treatment factor in the result section for each outcome. More importantly, the key take-away for journalists from this three level factor is that our results suggest international comparison news is more beneficial if it compares policies, not just outcomes. We added some discussion of these implications in the discussion section. 

4) How these samples play an vital role in external validity?

As with any experiment using a convenience sample, external validity is a significant limitation. We have added further acknowledgment of this limitation in the methods and results sections, while noting that MTurk convenience samples do have the advantage of greater demographic diversity compared to undergraduate convenience samples.

---

## [Decision Letter · Decision Letter 1]

22 Feb 2022

PONE-D-20-26385R1The curve not taken: Effects of COVID-19 international comparison newsPLOS ONE

Dear Dr. Kim,

Thank you for submitting your manuscript to PLOS ONE. After careful consideration, we feel that it has merit but does not fully meet PLOS ONE’s publication criteria as it currently stands. Therefore, we invite you to submit a revised version of the manuscript that addresses the points raised during the review process. In addition to the reviewer's comments, please address the follow:- how was sample size determined? was a power analysis performed?- please report participant gender- did excluding outliers affect the results? If so, note differences in text or supporting material. If not, note that there were no differences.- in the results section, make sure hypotheses are labelled/referenced the same way as in the introduction. For example, H2a is referenced in the results section, but no such sub-hypothesis is provided in the introduction. - were the news stories pretested or evaluated to determine whether they differ in any way other than intended?

We look forward to receiving your revised manuscript.

Kind regards,

Natalie J. Shook

Academic Editor

PLOS ONE

Journal Requirements:

Reviewers' comments:

Reviewer's Responses to Questions

**Comments to the Author**

1. If the authors have adequately addressed your comments raised in a previous round of review and you feel that this manuscript is now acceptable for publication, you may indicate that here to bypass the “Comments to the Author” section, enter your conflict of interest statement in the “Confidential to Editor” section, and submit your "Accept" recommendation.

Reviewer #1: (No Response)

Reviewer #2: All comments have been addressed

2. Is the manuscript technically sound, and do the data support the conclusions?

Reviewer #1: Partly

Reviewer #2: Yes

3. Has the statistical analysis been performed appropriately and rigorously? 

Reviewer #1: I Don't Know

Reviewer #2: Yes

4. Have the authors made all data underlying the findings in their manuscript fully available?

Reviewer #1: (No Response)

Reviewer #2: Yes

5. Is the manuscript presented in an intelligible fashion and written in standard English?

Reviewer #1: Yes

Reviewer #2: Yes

6. Review Comments to the Author

Reviewer #1: Interesting attempt. Here are a few comments/questions.

1. One might say that the overall result is overly common-sensical and trivial. It would have been much better, if more nuanced observations could be made. For instance, what are the relationships, if any, among different hypotheticals that the authors posed.

2. The authors mention in the conclusion section, as an illustrative example, the controversy surrounding gun control. In a similar vein as the reviewer's point above , it would have been more intriguing if the authors could make delineated observations and discussions as to what the similarities are and what the differences are between "Covid-19" and "gun control."

Reviewer #2: (No Response)

7. PLOS authors have the option to publish the peer review history of their article (what does this mean?). If published, this will include your full peer review and any attached files.

Reviewer #1: No

Reviewer #2: No

---

## [Author Response · Author response to Decision Letter 1]

5 Mar 2022

Response to Reviewers

The curve not taken

Due April 8, 2022

Highlight edited text in yellow

We thank you for inviting a second round of minor revisions on this manuscript originally submitted in August 2020. We have again responded to all of the reviewers’ concerns. We appreciate your service and attention to our work, and we humbly request that the remaining process for this timely manuscript be expedited as much as possible. Please see below to respond to the comments point by point. 

Editor 1. How was sample size determined? Was a power analysis performed?

⟶ Our sample size decision was based on available resources and a rule of thumb for media effects experiments of 100 participants per cell. Retrospectively we ran a power analysis based on an effect size of Cohen’s f = .14 from a solutions journalism experiment [1], resulting in a required sample size of 791. However, we were not aware of the solutions journalism literature when planning the study. In retrospect, we now view our independent variable international policy comparison news as a subset of solutions journalism, so we now acknowledge that connection in the literature review and discussion. 

Editor 2. Please report participant gender

⟶ Thank you for pointing out the reporting issue. Participant gender is reported on page 8 as follows. “We recruited a convenience sample of U.S. adults (final N = 848; 59.0% male, 40.4% female, 0.6% other) from Amazon’s Mechanical Turk (MTurk)”

Editor 3. Did excluding outliers affect the results? If so, note differences in text or supporting material. If not, note that there were no differences.

⟶ We appreciate your suggestion. Excluding outliers did not affect the result. Overall, it slightly increases effect sizes. We added the sentence in the result section. 

Editor 4. In the results section, make sure hypotheses are labelled/referenced the same way as in the introduction. For example, H2a is referenced in the results section, but no such sub-hypothesis is provided in the introduction. 

⟶ Thank you for pointing out our mistakes. Hypothesis labels have been corrected in the introduction and result sections: knowledge for testing & tracing (H1a), lockdown misperception (H1b), policy similarity misperception (H1c), support for policies successful in the comparison country (H2), one for presidential blame (H3), and one for trust in health experts (H4).

Editor 5. Were the news stories pretested or evaluated to determine whether they differ in any way other than intended?

⟶ Thank you for pointing out the manipulation check part. No manipulation check was included, although H1c (perceived policy differences between the two countries) is such a proximate outcome that it could be interpreted as a manipulation check. A manipulation check would have been useful only if effects on outcomes were not significant, to determine whether this was due to a failure of theory or a failure of stimulus delivery. Significant effects were found on multiple outcome variables. The treatment stories were not pretested, but we believe they were unusually well constructed to minimize treatment confounds. The stories had identical word counts across conditions and were presented in an identical layout with the appearance of real Wall Street Journal story webpages. After careful review, our assessment is that the stories do not differ in any important respects such as believability as a genuine and timely Wall Street Journal article, reading difficulty, style, length, or topic. If you see a specific potential confound in the stimuli, please let us know so we can acknowledge it in the discussion. 

Reviewer #1_1. One might say that the overall result is overly common-sensical and trivial. It would have been much better, if more nuanced observations could be made. For instance, what are the relationships, if any, among different hypotheticals that the authors posed.

⟶ Thank you for the comments. Although it may look obvious to some readers, international comparison news effect research has been limited not only in mass communication field but also in other fields. As discussed in the literature and discussion sessions, several decades of newsroom cost-cutting had resulted in major reductions in international newsgathering resources at U.S. news outlets. As a result, a few international news coverages focus on outcome comparison instead of policy comparison. As it is not a major form of international news, scholars’ interest in international policy comparison news remains undiscovered. The results in this study showed that international policy comparison news is a promising form of news to contribute to solving social issues. News audiences also want to learn policy comparisons across countries [2]. We hope that this study will encourage news outlets and scholars to pay attention to the merit of international policy comparison news. 

Reviewer #1_2. The authors mention in the conclusion section, as an illustrative example, the controversy surrounding gun control. In a similar vein as the reviewer’s point above, it would have been more intriguing if the authors could make delineated observations and discussions as to what the similarities are and what the differences are between “Covid-19” and “gun control.”

⟶ Thank you for the suggestion. International policy comparison news would be helpful to inform citizens, and may contribute to solving prolonged issues, such as gun control. One of the biggest differences would be the partisan level to Covid-19 and control issue. To be specific, Covid-19 originated relatively recently, and it would be expected to come to an end in the near future regardless of the current political debates about Covid-19 policies. However, the gun control issue is a strong partisan issue [3]. Throughout the long history of the debate over gun control in the U.S., those on either side of the political aisle have solidified their stances to a greater degree over time [4]. Thus, it would be more challenging to change people’s attitudes and policy stance about the gun issue compared to Covid-19. As your suggestion, it would be more interesting to discuss other important topics. However, this study focuses on Covid-19 issues for clear and concise writing. We plan on dealing with other issues in the subsequent research papers.

References

1. McIntyre K. Solutions journalism: The effects of including solution information in news stories about social problems. Journalism Practice. 2019;13(8):1029-33. doi: 10.1080/17512786.2019.1640632.

2. Pew Research Center. Americans give higher ratings to South Korea and Germany than U.S. for dealing with Coronavirus 2020. Available from: https://www.pewresearch.org/global/2020/05/21/americans-give-higher-ratings-to-south-korea-and-germany-than-u-s-for-dealing-with-coronavirus/.

3. Frankovic K. Republicans and Democrats agree on some gun limits 2018. Available from: https://today.yougov.com/topics/politics/articles-reports/2018/08/02/republicans-and-democrats-agree-some-gun-limits.

4. McGinty EE, Wolfson JA, Sell TK, Webster DW. Common sense or gun control? Political communication and news media framing of firearm sale background checks after Newtown. Journal of Health Politics, Policy and Law. 2016;41(1):3-40. doi: 10.1215/03616878-3445592.

---

## [Editor Report · Decision Letter 2]

30 May 2022

PONE-D-20-26385R2The curve not taken: Effects of COVID-19 international comparison newsPLOS ONE

Dear Dr. Kim,

Thank you for submitting your manuscript to PLOS ONE. After careful consideration, we feel that it has merit but does not fully meet PLOS ONE’s publication criteria as it currently stands. Therefore, we invite you to submit a revised version of the manuscript that addresses the points raised during the review process.

Thank you for your thoughtful responses to questions raised with the last review. For transparency and best practices, please report in text how sample size was determined. Also, report in text the results of the sensitivity analysis to demonstrate was size of effect your sample was powered to detect. 

We look forward to receiving your revised manuscript.

Kind regards,

Natalie J. Shook

Academic Editor

PLOS ONE
---

## [Author Response · Author response to Decision Letter 2]

31 May 2022

We thank you for inviting a third round of minor revisions on this manuscript. We have reported a) how sample size was determined and b) the results of the sensitivity analysis to demonstrate the size of effect this study's sample was powered to detect. Please see below to respond to the comments point by point. 

Editor 1. How please report in text how sample size was determined

⟶ As we responded in the second revision, we reported how we decided our sample size in the method section as follows, "The sample size decision was based on available resources and a rule of thumb for media effects experiments of 50 participants per cell [1]."

Editor 2. Report in text the results of the sensitivity analysis to demonstrate the size of effect your sample was powered to detect. 

⟶ Thank you for pointing out the missing information in the text. We added the power analysis result in the method section as follows, "Retrospectively, we ran a power analysis based on an effect size of Cohen’s f = .14 from a solutions journalism experiment [2] and .95 power using G*power application, resulting in a required sample size of 792. The retrospective power analysis confirmed that our sample size is appropriate."

---

## [Editor Report · Decision Letter 3]

23 Jun 2022

The curve not taken: Effects of COVID-19 international comparison news

PONE-D-20-26385R3

Dear Dr. Kim,

We’re pleased to inform you that your manuscript has been judged scientifically suitable for publication and will be formally accepted for publication once it meets all outstanding technical requirements.

Kind regards,

Natalie J. Shook

Academic Editor

PLOS ONE
---

## [Editor Report · Acceptance letter]

2 Aug 2022

PONE-D-20-26385R3 

The curve not taken:
Effects of COVID-19 international comparison news

Dear Dr. Kim:

I'm pleased to inform you that your manuscript has been deemed suitable for publication in PLOS ONE. Congratulations! Your manuscript is now with our production department. 

Kind regards, 

on behalf of

Dr. Natalie J. Shook 

Academic Editor

PLOS ONE